# The Inhibition of Vessel Co-Option as an Emerging Strategy for Cancer Therapy

**DOI:** 10.3390/ijms25020921

**Published:** 2024-01-11

**Authors:** Iván Carrera-Aguado, Laura Marcos-Zazo, Patricia Carrancio-Salán, Elena Guerra-Paes, Fernando Sánchez-Juanes, José M. Muñoz-Félix

**Affiliations:** 1Departamento de Bioquímica y Biología Molecular, Universidad de Salamanca, 37007 Salamanca, Spain; icasdp@usal.es (I.C.-A.); lauramarcos@usal.es (L.M.-Z.); patriciacs@usal.es (P.C.-S.); elegpaes98@usal.es (E.G.-P.); fsjuanes@usal.es (F.S.-J.); 2Instituto de Investigación Biomédica de Salamanca (IBSAL), 37007 Salamanca, Spain

**Keywords:** vessel co-option, angiogenesis, adhesion, extracellular matrix

## Abstract

Vessel co-option (VCO) is a non-angiogenic mechanism of vascularization that has been associated to anti-angiogenic therapy. In VCO, cancer cells hijack the pre-existing blood vessels and use them to obtain oxygen and nutrients and invade adjacent tissue. Multiple primary tumors and metastases undergo VCO in highly vascularized tissues such as the lungs, liver or brain. VCO has been associated with a worse prognosis. The cellular and molecular mechanisms that undergo VCO are poorly understood. Recent studies have demonstrated that co-opted vessels show a quiescent phenotype in contrast to angiogenic tumor blood vessels. On the other hand, it is believed that during VCO, cancer cells are adhered to basement membrane from pre-existing blood vessels by using integrins, show enhanced motility and a mesenchymal phenotype. Other components of the tumor microenvironment (TME) such as extracellular matrix, immune cells or extracellular vesicles play important roles in vessel co-option maintenance. There are no strategies to inhibit VCO, and thus, to eliminate resistance to anti-angiogenic therapy. This review summarizes all the molecular mechanisms involved in vessel co-option analyzing the possible therapeutic strategies to inhibit this process.

## 1. Tumor Vascularization

Angiogenesis is a process that consists of the formation of new blood vessels from pre-existing ones to create a fine and mature vascular network [1,2]. New blood vessels emerge from new branches of the original vessels in a process called sprouting angiogenesis [2,3]. Sprouting angiogenesis is a physiological process that depends on the balance between pro-angiogenic and anti-angiogenic growth factors and cytokines. In normal physiological conditions, anti-angiogenic factors are predominant promoting endothelial cell quiescence. However, cancer cells can secrete numerous pro-angiogenic growth factors that induce angiogenic switch, and then, the formation of new blood vessels [4].

During the first stages of cancer growth, small tumors develop avascular growth and take nutrients from extracellular space by diffusion [1]. Nevertheless, when tumor volume exceeds 2 mm^3^, the uptake of nutrients by diffusion is not enough to maintain tumor growth and cancer cells undergo hypoxia [3]. Under these conditions, the proteasome degradation of hypoxia inducible factor 1β (HIF-1β) is inhibited and translocates into the nucleus to form a heterodimeric complex together with HIF-1β. This complex binds the DNA hypoxia response elements (HRE) and induces the expression of pro-angiogenic molecules such as vascular endothelial growth factor VEGF, metalloproteinases or angiopoietins [1,5]. Moreover, the metabolism from hypoxic tumor cells produce molecules that acidify tumor microenvironment and acidosis can stimulate the production of pro-angiogenic factors [1]. The most important cytokine secreted by tumor cells is vascular endothelial growth factor (VEGF-A). VEGF interacts with its receptor VEGFR2, present in endothelial cells, and stimulates its tyrosine kinase activity, so that it autophosphorylates and triggers numerous intracellular signaling pathways (PI3K/AKT, MEK/ERK, MAPK) that promote endothelial cell (EC) proliferation, EC migration and EC survival to create new blood vessels [6,7,8]. EC migration is favored by matrix metalloproteinases (MMPs), which degrade the extracellular matrix to facilitate cell movement into the tumor. In this way, simultaneous migration and division allow ECs to organize themselves to originate the sprouts and branches that will give rise to the new blood vessels allowing the penetration and, in consequence, vascularization of the tumor [7,9]. Finally, the newly formed vascular network is stabilized by several molecules such as angiopoietin 1 (Ang-1) [10]. Ang1 promotes the interaction between ECs and pericytes [11]. However, these angiogenic tumor vessels do not become fully mature, since the pericyte coverage is partial and the basal lamina is not fully developed. As a result, they have a high permeability and are not fully functional. This fact hinders the influx of antitumor therapies, which renders them ineffective (Figure 1) [6,12].

## 2. Angiogenesis vs. Vessel Co-Option

Vessel co-option (VCO), also called vascular co-option, angiotropism or pericytic mimetism, is a non-angiogenic mechanism by which cancer cells migrate through the abluminal side of vessels, displacing non-tumoral cells and recruiting immune cells such as neutrophils or M1 macrophages until the interaction between pre-existing blood vessels and tissular parenchyma takes place. They are thus taking benefit of oxygen and nutrients supply without the necessity of the formation of new blood vessels [2,14,15]. In contrast to the initial ideas that suggested that in the early stages of neoplasias a transdifferentiation from non-angiogenic tumor growth towards a neo-angiogenic vascularization was necessary, it has since been demonstrated that tumors can grow exclusively in non-angiogenic growth patterns, even in macroscopic lesions [16]. Most of the molecular mechanisms that control non-angiogenic cancer growth are still unknown.

However, it is accepted now that tumor vascularization mechanisms comprise a set of dynamic spatial and temporal processes mediated by the tumor microenvironment, which allows us to clarify some ideas [15]:(i)Although there are tumors undergoing only angiogenesis, most of them show both types of vascularization, non-angiogenic and angiogenic, even within the same lesion [16,17,18].(ii)Tumors can modify their growth pattern depending on their stage or the different features of the neoplasia, the origin of the primary tumor and the location of the pre-metastatic niche [19].(iii)Irregular growth pattern based in VCO involves several complications during the surgical removal of the tumor and resistance to chemotherapy [14,20].(iv)Transdifferentiation from the angiogenic to non-angiogenic phenotype has been described as a response to therapeutic selective pressure, being proposed as a major resistance to anti-angiogenic therapy (AAT) mechanism both intrinsic, from the beginning of treatment, and acquired, observed after an initial response and being able to return to the angiogenic process when therapy is interrupted [14,17].(v)VCO has been related with the metastatic cascade (mobility, cell adhesion and invasiveness), plasticity, latency and tumor cell survival, being associated with epithelial-to-mesenchymal (EMT) transition and paracrine interaction between cancer cells and pericytes or tumor cells from co-opted vessels [21,22].

## 3. VCO as a Resistance Mechanism to Anti-Angiogenic Therapy

VCO was first identified in non-small cell lung cancer (NSCLC) in 1997 by Pezzella, et al., who proposed that this strategy of tumor growth was a mechanism of resistance to angiogenic therapy [23,24]. In the following years, it was accepted that tumors do not grow using a single vascularization mechanism, but microvascular growth patterns can vary in time and space. Thus, tumors can initially grow undergoing angiogenic patterns and then develop VCO as a resistance mechanism to therapy [24].

Numerous studies were conducted with liver metastases, hepatocellular carcinoma [22], glioblastoma [25], lymph node metastases [26] or brain metastases [27]. All of them have generated similar results, in which the absence of response to antiangiogenic therapy appears fundamentally in tumors that perform VCO. The fact that these observations have been made in cancers of a very diverse nature leads to the confirmation of the hypothesis put forward by Pezzella et al. However, the mechanisms by which VCO allows evasion of AAT are not yet known [24,28,29]. It is easy to understand that given that AAT was designed to prevent the formation of new vessels, it is expected not to show any effect on pre-existing vessels. For this reason, after AAT, VCO tumors will have a blood supply available that will allow it to continue growing via VCO [24].

### 3.1. Histological Features of VCO vs. Angiogenic Tumors

Through the years, scientists have classified primary lung tumors and metastasis according to their growth patterns. In lungs, non-angiogenic tumors that grow via VCO can be differentiated from angiogenic tumors because of their growth preserving the normal lung architecture, allowing the co-option of the blood vessels in the alveolar wall. The opposite is true for angiogenic tumors, which grow by destroying the normal structure of the wall.

In lungs, five non-angiogenic growth patterns have been described to date: alveolar, lepidic, papillar, interstitial and perivascular cuffing. The alveolar growth pattern was the first pattern described by Pezzella et al. in NSCLC in 1997 [23], but there are reports from the 1860s describing tumor cells growing inside the alveoli and filling the alveolar space displaying a ‘bird’s nest-like’ formation, although they did not relate this to VCO [30]. In this pattern, tumor cells grow within the alveoli, completely filling the alveolar spaces (Figure 2). The alveolar wall is incorporated into the tumor structure, so vessels can be co-opted. Remarkably, the vascular network within the tumor closely resembled that of a healthy lung, with cancer cells co-opting these vessels by growing within the alveolar air spaces. Pneumocytes can be observed within the tumor compartment but a loss of these pneumocytes can be observed from the periphery to the center of the metastasis, but the vessels remain intact and retained within the tumor [22,31]. In the lepidic growth pattern tumor cells grow within the alveoli spreading over the surface of the alveolar walls in a thin layer gradually replacing pneumocytes, which grow adhered to the basement membrane and do not fill the alveolar spaces completely [14,32]. The papillar growth pattern is associated with the lepidic growth pattern and is exclusive of lung adenocarcinomas. This pattern is non-angiogenic but can present an angiogenic component as cancer cells co-opting the alveolar capillaries can induce the formation of new blood vessels [14,31]. In the interstitial growth pattern cancer cells grow within the lung interstitium, thickening the alveolar walls. Nonetheless, they can still hijack the capillaries of these walls. The expansion of the cancer cell population leads to the widening of alveolar walls, yet the majority of alveolar spaces are preserved. However, there is evidence of invasion into the alveolar air spaces at the center of some metastases [31,33]. In the perivascular cuffing pattern tumor cells grow surrounding larger blood vessels, like a cuff around them and without additional blood vessels [2,21]. As stated above, co-opted vessels are pre-existent. This fact can be known because the tumor and the normal pulmonary vasculature present the same phenotype and architectural pattern. These vessels express LH39, a marker for the alveolar wall basement membrane and are negative for α_v_β3 integrin, a marker for angiogenesis [34]. Moreover, the endothelial cell proliferation rate is lower compared to angiogenic tumors and are very similar to the rate in the normal tissue [35,36].

On the other hand, four angiogenic growth patterns have been described: destructive, basal, diffuse and pushing, characterized by the replacement of lung parenchyma by a desmoplastic process, and the presence of a disorganized vasculature (Figure 2) [14]. This desmoplastic reaction is absent or minimal in tumors growing via VCO [21] but C38 (a colorectal cancer cell line) lung metastasis can induce desmoplastic responses, replacing the alveolar walls into intratumoral tissue columns that are not infiltrated by tumor cells [36].

### 3.2. VCO Induced Resistance in Lung Cancer and Lung Metastases

Although it has been suggested that VCO is a mechanism that takes place in solid tumors, this phenomenon has been found more frequently in highly vascularized organs such as the brain, liver and lungs [2,14]. In fact, this phenomenon in the lungs was first described by Pezzella et al. in the 1960s in a study using 500 samples from patients with NSCLC in primary and metastatic tumors [23]. In this research study, a growth pattern in which cancer cells filled the alveolar spaces without the destruction of the normal lung parenchyma by co-opting the septal capillaries was observed. Years later, this identified growth pattern was named ‘alveolar’ [16]. The alveolar growth pattern has been associated to the periphery of squamous cells carcinoma (SqCCs) and solid adenocarcinoma, although it has been described in all the histologic types [35]. In a recent study, authors looked at the different microvascular growth patterns in 553 patients with NSCLC. They identified non-angiogenic growth patterns in the 18% of the samples, being the 82% with angiogenic growth pattern (basal, diffuse or papillary). They found that non-angiogenic alveolar tumors showed poor prognosis regardless of the tumor microenvironment, evaluated by Glut1, CD3, CD8, CD45 or PD-1) [37].

In the context of pulmonary metastases, Bridgeman et al. demonstrated that breast cancer cells from the 4T1 cell line metastasize in lungs undergoing VCO in alveolar or interstitial growth patterns [16]. These 4T1 lung metastases are resistant to sunitinib-based therapy. However, renal cancer cells from the RENCA cell line metastasize in lungs undergoing angiogenesis in the 80% of the lesions and are sensitive to sunitinib, a VEGFR2 inhibitor [16]. This first study demonstrated that VCO growing lung metastases are resistant to anti-angiogenic therapy whereas angiogenic lung metastases respond to anti-angiogenic therapy. More recently, it was confirmed that chronic inhibition with sunitinib promoted VCO in lung metastases from RENCA tumor cells [38].

The association of vessel VCO with resistance mechanisms has also been described in other organs. In the liver, using an experimental model of hepatocellular carcinoma (HCC) acquired a resistance to sorafenib, a TKI inhibitor, and was associated with increased VCO [22].

### 3.3. VCO Induced Resistance in Brain Tumors and Brain Metastases

VCO has also been identified in primary tumors of the central nervous system (gliomas) and in metastatic brain lesions produced by cells from colorectal, breast, lung and skin cancers [24,39,40]. Two VCO growth patterns have been described where cancer cells grew surrounding pre-existing blood vessels in the tissue: the perivascular growth pattern (also known as perivascular cuffing) and the diffuse infiltrating growth pattern [2]. The main difference between the two growth patterns lies in the fact that in the perivascular growth pattern, cancer cells can adhere to the vessel surface and disrupt the interaction between the blood vessels with pericytes and astrocytes, contributing to destroying the integrity of the blood–brain barrier [24]. However, in the diffuse infiltrating growth pattern, tumor cells invade the parenchyma without establishing a permanent interaction with the blood vessels, generating little disruption of the blood–brain barrier [30].

Non-angiogenic growth pattern in the brain can be established due to the absence of connective tissue in brain parenchyma and the presence of a fine network that contains numerous extracellular matrix proteins such as laminin, collagens and fibronectin in its basal membrane [30,41]. In vivo studies have demonstrated the involvement of several molecules in VCO tumor growth such as Bradikinin-2 receptor, chemoatractant system SDF1/CXCR4, Wnt7, the mutated receptor of EGFR (EGFRvIII) and ephrin-B2 [42,43,44,45,46,47]. However, in brain metastases, this mechanism is regulated by two adhesion proteins expressed by tumor cells: L1CAM and β1 integrin [39,44,48,49]. Despite the tumor cells’ own intrinsic ability to co-opt vessels in brain tissue, anti-angiogenic therapy with cediranib, sunitinib and vandetanib, inhibitors of receptors involved in the process of angiogenesis, has been reported to promote the expansion of cancer cells along pre-existing blood vessels in different types of gliomas. Moreover, antibodies against VEGF, favor the expansion of cancer cells along pre-existing blood vessels in different types of gliomas and brain metastases [27,50,51,52,53].

Tumors undergoing VCO in brain tissue show intrinsic resistance to AAT because their growth does not depend on angiogenic factors targeted by this type of treatment [54,55]. For this reason, it is thought that the preclinical and clinical results have been very limited with respect to what was expected, especially when using the monoclonal antibody bevacizumab [56,57,58,59]. Furthermore, several studies have shown that the use of anti-angiogenic agents in brain tumors induces a higher invasive ability of cancer cells, allowing them to invade the adjacent brain tissue and even metastasize to other distant locations [55,60,61,62]. This is due to the fact that VEGF suppresses cell motility and, thus, its inhibition induces tumor cells to acquire a mesenchymal phenotype that favors their intravasation [60,63]. On the other hand, regarding the microenvironment, it has been described that the long-term use of AAT generates the presence of immunosuppressive macrophages that, in turn, limit an adequate response to treatments [64].

Given the clinical resistance induced by VCO growth in gliomas and brain metastases, several strategies are currently being developed to prevent cells from growing in a non-angiogenic growth pattern. For example, inhibition of the interaction of cancer cells with pericytes by suppressing CD44 and CDC42 could induce cancer cells to develop another tumor growth pattern and has enhanced the anti-tumor immune response [65,66]. VCO growth in the brain can also be prevented by targeting L1CAM, β1-integrin or CXCR4 molecules, as we will detail in the next section. Inhibition of CXCR4 reduces tumor invasion in glioblastoma cells and increases their sensitivity to radiotherapeutic agents while suppression of β1-integrin could favor cells not to settle around pre-existing blood vessels [67]. Finally, a mathematical model of glioma has suggested that a reduction in hypoxia induced by non-functional VCO could favor the presence of M1 macrophages and inflammatory cytokines to enhance the regression of tumors with vascular co-option [51,54]. However, further research on the mechanisms and molecules that drive VCO in the brain is needed to develop truly effective treatments and strategies to avoid or reduce current resistance.

### 3.4. VCO Induced Resistance in Hepatocarcinoma and Liver Metastases

In 2017, a consensus was established to identify the histopathological growth patterns (HGP) in liver metastases [68]. According to this criteria, three HGP exist in liver metastases, characterized by different mechanisms of vascularization, desmoplastic HGP (dHGP), replacement HGP (rHGP) and pushing HGP (pHGP) [15,68], although primary tumors also grow undergoing similar vascularization growth patterns [28]. In the dHPG pattern, tumor cells are physically separated from healthy liver parenchyma by a fibrotic structure of stromal tissue [68,69,70]. In this growth pattern, cancer cells use sprouting angiogenesis as a mechanism of vascularization [70,71]. On the other hand, in the rHGP pattern, tumor cells replace hepatocytes that are present in the healthy liver–hepatic interface [69,70] and thus gaining access to the sinusoidal capillaries of the liver for VCO [35], thus maintaining the normal tissue structure [44,58]. Finally, in the pHGP pattern, cancer cells directly contact the liver tissue, compressing it, and thus altering its structure [47]. These growth patterns have been identified in several patients, especially in those with colorectal cancer liver metastases (CRLM) [72]. In these studies, it has been observed that the dHGP pattern is associated with increased survival and prognosis in both metastatic tumors and hepatocellular carcinoma [65,72]. For example, Meyer et al. analyzed samples of liver metastases from different origins (colon, breast, melanoma, etc.) and observed that 5-year survival, overall survival and post-operative relapse-free time were higher in metastases with dHGP patterns compared to those with rHGP patterns. In another study, Fleischer et al. studied liver metastases of colorectal cancer and found that tumors with a dHGP pattern were associated with better prognosis than tumors with rHGP and pHGP patterns [71]. These differences could lie in the different types of tumor microenvironments established in each histopathological growth pattern. In previous studies, the higher infiltration of immune cells, especially CD8+ T lymphocytes, has been observed, especially in desmoplastic tumors (dHGP) in comparison with replacement pattern tumors (rHGP) [15]. This phenomenon could lead to an antitumor tumor microenvironment (TME) that would favor the destruction of tumor cells in the desmoplastic pattern, improving the prognosis.

The use of AAT in liver cancer began in 2004 with the approval of the monoclonal antibody bevacizumab for co-administration with chemotherapy in the treatment of liver metastases from colorectal cancer [73]. Since then, five antiangiogenic drugs have been approved to treat hepatocellular carcinoma (sorafenib, lenvatinib, cabozantinib, ramucirumab and regorafenib), as have another two antiangiogenic compounds for the treatment of liver metastases (aflibercept and regorafenib) [15]. However, this therapy has not shown beneficial effects in patients, since no improvements in overall survival or disease-free progression have been observed [30]. One of the reasons for the lack of therapeutic response in both primary and metastatic tumors is that VCO can act as a mechanism of resistance (acquired and/or intrinsic) to antiangiogenic therapy [14,24,70,73]. Several studies conducted in recent years with samples from liver tumors have reaffirmed this idea [14]. For example, Kuczynski et al. developed a preclinical model of hepatocellular carcinoma and applied a treatment based on sorafenib, which showed the capacity to prevent the formation of new tumor blood vessels [22]. However, some of the tumors acquired resistance to the drug, which was manifested by increased invasiveness that allowed the tumor cells to gain access to pre-existing sinusoidal blood vessels for VCO. Therefore, these results indicated that the lack of response was due to vascular co-option [22]. On the other hand, Frentzas et al. correlated the lack of response to antiangiogenic therapy with the histopathological growth pattern of colorectal cancer liver metastases from patients who had been treated with bevacizumab and chemotherapy prior to surgery. In this study, they observed that patients who responded adequately to treatment had dHGP-type metastases, i.e., angiogenic. On the other hand, those patients who did not show this response had metastases with an rHGP pattern, i.e., vessel co-option dependent [17]. This study also analyzed the metastases of patients who suffered relapses of the disease after applying bevacizumab and observed that most of them had developed by VCO metastases (rHGP) [21]. Thus, this work demonstrated that the use of antiangiogenic therapy can promote tumor growth in VCO. Finally, Frentzas et al. generated a murine model of advanced liver metastasis using the HT29 cell line and treated the mice with an anti-VEGF antibody. This antibody showed the ability to reduce the growth of dHGP metastases, but not that of rHGP patterned tumors [17,29]. Subsequently, Lazaris et al. found that the anti-angiogenic drug bevacizumab reduced the number of blood vessels in colorectal cancer liver metastases developing in a desmoplastic pattern (dHGP). In contrast, this decrease was not observed in tumors with a replacement pattern (rHGP) [74].

In addition to being responsible for resistance to AAT, VCO could also be involved in resistance to chemotherapy in liver cancer as a consequence of the microenvironment that is established in rHGP-type tumors [70]. For example, Lu et al. observed that, in rHGP tumors, endothelial cells produce the ligand Jagged-1, which induces a stem cell phenotype in tumor cells [75] which could affect the efficacy of chemotherapy [70]. However, this aspect is still under investigation and further work is needed.

## 4. Cellular and Molecular Mechanisms Involved in VCO

There is little information about which conditions of the TME induce and facilitate vessel co-option and about the molecular mechanisms involved in cancer cells and tumor microenvironment behavior. In this section, we summarize some of the conclusions derived from these studies (Table 1).

### 4.1. Different Blood Vessels

It is not difficult to understand that blood vessels that are co-opted by tumor cells are more similar to normal blood vessels than angiogenic blood vessels, which, in theory, are more leaky. This has been recently demonstrated in a study using single cell analysis, confirming that cells present in co-opted blood vessels are similar cells belonging to blood vessels from healthy tissue at a molecular level, whereas angiogenic ones are not [38].

Regarding endothelial cells (ECs), one of the main differences found between ECs from angiogenic and non-angiogenic tumors is the presence of proliferative ECs, immature ECs and tip cells in angiogenic tumors; whereas in non-angiogenic tumors, which develop by VCO, these cells are practically nonexistent, showing quiescent EC signatures. Furthermore, after analyzing the transcriptomes of the ECs present in both tumor types, co-opted ECs were very similar to ECs present in healthy lung tissue [38].

These results were first observed in metastasis models generated with RENCA cells, which are characterized by both growth patterns (angiogenic-80% and VCO-20%). In the control groups not treated with AAT (sunitinib), immature, proliferative ECs and tip cells predominated, whereas in the groups treated with sunitinib these cells disappeared. In metastases models generated with 4T1 cells that were characterized by tumors that grow by vessel co-option in a natural way, these characteristic angiogenic ECs did not appear, and instead, quiescent ECs predominated [38].

In the same study, the authors also characterized the pericytes present in angiogenic and VCO tumors, identifying two types of pericytes: ‘angiogenic pericytes’ that appeared in untreated metastases, and ‘quiescent co-opted pericytes’ that appeared in metastases treated with sunitinib and in metastases were generated with 4T1 cells (development by VCO). As with ECs, quiescent pericytes appearing in non-angiogenic tumors were transcriptionally similar to pericytes from healthy tissues and mainly expressed genes related to vasodilation and quiescence. On the other hand, angiogenic pericytes showed genes related to cell migration and motility processes and extracellular matrix reorganization [38].

#### Targeting Tumor Vasculature to Inhibit VCO

One of the first strategies to consider when inhibiting VCO is targeting the tumor vasculature. Thinking in pruning co-opted blood vessels, normalizing them or promoting tumor angiogenesis can be a good strategy for targeting VCO. The destruction of tumor vasculature can be considered a good strategy for blocking VCO. However, how can we destroy the pre-existing blood vessels? Targeting tumor angiogenesis has been considered the strategy for pruning vasculature although not all the tumors develop tumor angiogenesis [6].

Anti-angiogenic strategy emerged in the early 1970s when molecular mechanisms of tumor angiogenesis were described and acquired the name of antiangiogenic therapy [76,77]. This type of strategy is based on reducing the number of tumor blood vessels with the aim of depriving cancer cells of nutrients and thus slowing their growth [77,78].

The first anti-angiogenic drug approved for colorectal cancer metastases was bevacizumab, a humanized monoclonal antibody that interacts with VEGF to prevent the formation of new blood vessels [79,80,81]. Subsequently, other anti-angiogenic molecules have also been developed, such as sunitinib or sorafenib, which in this case inhibit the tyrosine kinase activity of TK receptors such as VEGFR2 [16]. Although the pre-clinical results were truly promising, currently the inhibition of the angiogenesis process as monotherapy has shown very limited benefits [82,83]. In turn, the reduction of blood vessels restricts the arrival of chemotherapeutic agents that use the bloodstream as a transit route to the tumor site, thus increasing resistance to treatment [84].

On the other hand, this type of therapy is based on the fact that all solid tumors require the formation of blood vessels in order to grow and progress [85]. However, as we are focusing in this review manuscript, some types of cancer such as breast, pancreatic, lung or prostate cancer show intrinsic resistance (from the start of therapy) due to their high vascularization, which allows the cells to use other mechanisms to grow [15].

Thus, is it possible to inhibit VCO with angiogenic therapy? The first issue to address is to know if these vessel-co-opting tumors produce VEGF. Some authors have addressed this question by hypothesizing that co-opted vessels in tumors with high levels of angiopoietin-2 undergo vessel regression and hypoxia. These higher levels of hypoxia would stimulate cancer cells to secrete VEGF in these ‘non-angiogenic’ tumors [86]. However, VEGF needs the presence of VEGFR2 (or other receptors) in ECs to exert its function. Recent studies have shown that co-opted ECs show a quiescent phenotype [38]. Moreover, there are some functional evidences that these tumors do not respond to anti-angiogenic therapy with sunitinib (inhibitor of tyrosine kinase activity of VEGFR2) [16,38] suggesting that there is no upregulation of VEGFR2 in co-opted vessels. However, little is known about other angiogenesis inhibitors.

Given these issues, perhaps AAT is not the best approach for targeting blood vessels in order to inhibit VCO. Given the problems described previously regarding co-opted blood vessels regression and hypoxia, we can consider vascular normalization as a promising strategy.

As a consequence of the limitations of AAT, the concept of vascular normalization emerged in 2001. Vascular normalization consists of partially reducing the number of blood vessels and favoring the functioning of the remaining vessels thanks to an improvement in pericyte coverage and in the junctions of endothelial cells with the basement membrane [87,88,89,90]. In this way, the tumor vessels become less permeable and have a homogeneous blood flow that allows adequate oxygen and nutrients to reach all the cancer cells that constitute the tumor [77]. In other words, the increase in blood flow reduces hypoxia and limits metastatic dissemination due to the corrections exerted on the vessel barrier. In addition, the delivery of antitumor drugs is favored and the cells respond to treatments [77].

On the other hand, it has been observed that the normalization of the vessels improves the infiltration of lymphocytes into the tumor, whose activation, in turn, promotes greater vessel normalization, generating a cycle of mutual regulation that contributes to fighting and destroying cancer cells [91]. This type of strategy has been described as being achieved by employing low doses of anti-angiogenic agents such as those mentioned above in combination with chemotherapy [81]. However, vessel normalization is difficult to apply in a clinical setting since it is necessary to find the correct dose of each antiangiogenic drug for each patient. Moreover, the effect on tumor vessels is not permanent, probably because the tumors become resistant to the anti-angiogenic molecules and use other mechanisms to generate defective blood vessels again [12,76].

As mentioned above, vessel normalization can be achieved by administering low doses of anti-angiogenics such as anti-VEGF or anti-Ang2 molecules [90], but a different method to normalize tumor vasculature and also tumor stroma is by administering losartan, which increases blood flow and also normalizes tumor stroma [92].

Wong et al. described in 2015 a new strategy called ‘vascular promotion’ which consists of the generation of a network of new blood vessels [77,82]. The rationale of vascular promotion strategy is the formation of new blood vessels and the increase of blood flow by using vasodilators such as verapamil to improve drug delivery while reducing tumor hypoxia. Thus, tumors can receive less chemotherapy, as the drug delivery is improved [6,12,78]. This was described in lung and pancreatic tumors by administering a low dose of cilengitide, an a_v_β_3_ inhibitor that behaves as pro-angiogenic when used at low doses [93], gemcitabine and verapamil, calcium channel blockers. The combination of low dose cilengitide, gemcitabine and verapamil reduced tumor growth and extended survival in tumors that showed higher blood vessel density and lower levels of tumor hypoxia [82]. Other strategies have been shown to induce ‘vascular promotion’ by using lysophosphatidic acid or eribulin [94,95]. The use of vascular promotion strategies can be an interesting strategy to consider for the inhibition of VCO. However, there are many challenges such as the acquisition of the angiogenic switch or the promotion of new and functional blood vessels.

Finally, with respect to tumor hypoxia and angiogenic pathways, it is necessary to mention that hypoxia markers (HIF1α, carbonic anhydrase 9 (CAIX) and Glut1) are regulated by MET (hepatocyte growth factor receptor) [63]. VEGF inhibition leads to the activation of MET, which correlates with poor prognosis [61,62,96]. Given that in VCO tumors, the VEGF/VEGFR2 axis may be downregulated, higher MET activation and consequent tumor hypoxia is expected to be seen (Figure 3).

### 4.2. Cancer Cell Adhesion

One of the mechanisms that control and promote VCO is the adhesion between cancer cells and the pre-existing blood vessels [2,14]. It is believed that tumor growth, tumor invasion and tumor expansion in VCO is mediated by the adhesion to pre-existing blood vessels (Figure 4). Considering that pre-existing blood vessels are covered by a mature basement membrane, cancer cells can be adhered to extracellular matrix components from the basement membrane or endothelial cells [29].

In brain tumors or brain metastases, cancer cell adhesion to pre-existing blood vessels is mediated by several integrins such as α_3_, α_6_ or β_1_. In a mouse model of brain metastases using C38, HT25 (murine colon carcinoma), h1650 (lung carcinoma), ZR75 (mammary carcinoma) and HT1080 (fibrosarcoma) cell lines, it was detected α_3_ integrin as a protein mediating cell adhesion between cancer cells and pre-existing blood vessels [48].

In acute lymphoblastic leukemia (ALL) cells migrate and invade the central nervous system by using the interaction between cancer cell α_6_ integrin and laminin from pre-existing blood vessels, this process being mediated by PI3K [97].

In glioblastomas, β_1_ integrin was found to be upregulated in bevacizumab resistant patients. This protein was identified to mediate cancer cell adhesion to the vascular basal membrane. Xenograft models of glioma demonstrated that inhibition of β1 integrin with the antibody OS2966 reduced tumor growth and metastases incidence [49,98].

The cell adhesion molecule L1 (L1CAM) is a transmembrane glycoprotein present in the cell membrane and is expressed in the nervous system during development, but its expression is altered in different types of cancer and is linked to worse outcomes of the disease [41]. Interestingly, L1CAM presents a dual role depending on the cell type in which it is expressed. Its expression in normal cells increases adhesion and response to therapy while in cancer or transformed cells it promotes epithelial mesenchymal transition, motility, migration and increases invasion and therapy resistance [99,100]. It is also believed to be necessary for tumor cells to develop VCO in brain metastases as it has been shown that its depletion prevents VCO from taking place in the brain [41,44]. RNAi-knockdown of L1CAM in breast cancer cells suppresses invasion and migration. L1CAM is a known mediator of VCO in pericytes that reinforces de β_1_-integrin/ILK signaling pathway [41] but it also mediates the spread of cancer cells into the vasculature and the interaction between cancer cells [44]. The interaction between the tumor and endothelial cells is key to the development of VCO. In brain metastases, the expression of L1CAM is increased in the interface between tumor and endothelial cells [101]. The disruption of L1CAM in cancer cells resembles the impact of targeting β_1_-integrin, impeding the cancer cells to spread on co-opted vessels, thus affecting proliferation and preventing the formation of macrometastases [41].

The role of these integrins α_3_, α_6_ or β_1_ or L1CAM have been described in the cancer types and studies in these experimental models described above. However, it is very easy to understand that co-opted vessels show a mature basement membrane, covered by ECM proteins such as fibronectin, laminin or collagen IV and tenascin C. The presence of this mature basement membrane can favor cancer cells to adhere to and co-opt these blood vessels. For this reason, the inhibition of cancer cell adhesion to pre-existing blood vessels can be a promising strategy to inhibit vessel co-option.

#### Targeting Cancer Cell Adhesion to Inhibit VCO

The inhibition of these integrins to avoid and prevent cancer growth has been studied for years focusing on the tumoral compartment or with the purpose of inhibiting tumor angiogenesis but not with the intention of VCO inhibition. Studies with human breast cancer cell lines showed that the inhibition of β_1_ integrin with AIIB2, an antibody against the extracellular domain of β_1_ integrin, promotes breast cancer cell apoptosis and reduces tumor growth in vivo. β_1_ integrin inhibitory antibody induces apoptosis of breast cancer cells, inhibits growth and distinguishes malignant from normal phenotype in three-dimensional cultures and in vivo [102]. Moreover, using this AIIB2, this research team found an increase in radiotherapy efficacy in an AKT-dependent mechanism by using β_1_ integrin inhibition [103]. AIIB2 also reduced invasiveness and enhanced radioresponse in a xenograft model of MCF-7 breast cancer β_1_ integrin via NF-κB signaling is essential for acquisition of invasiveness in a model of radiation treated in situ breast cancer [104].

As we mentioned before, the inhibition of β_1_ integrin with the antibody OS2966 potentiated the anti-tumor effect of bevacizumab in an experimental model of bevacizumab-resistant glioblastoma [49]. ATN-161 (Ac-PHSCN-NH2) is a pentapeptide derived from the PHSRN sequence of fibronectin. By mimicking this sequence, binding to fibronectin is not possible for β_1_ integrin, and thus its action is blocked. ATN-161 demonstrated the inhibition tumor growth and metastases in experimental models using MDA-MB-231 breast cancer cells [105]. In these cells, ATN-161 reduced tumor growth and the incidence of skeletal metastases and metastases in the lungs, liver and spleen [105]. This was observed using tumor cell models. Moreover, its role on endothelial cells and angiogenesis was elucidated by using in vitro models of tube formation and in vivo using choroidal neovascularization models [106]. ATN-161 also demonstrated its benefits when used in combination with chemotherapy. In an experimental model of colorectal cancer liver metastasis using the cell line CT26, the administration of 5-FU in combination with ATN-161 reduced the number of liver metastases and liver weight. Moreover, it demonstrated a reduction of blood vessel density and an increase in apoptosis in these metastases treated with the double combination [107]. However, these authors did not study in detail the vascularization growth patterns on these liver metastases, which has been described to undergo VCO, at least in lung tissue [16], and this effect of ATN-161 cannot be attributed to an effect on VCO decrease. In 2006, in a phase I clinical trial, ATN-161 was used in patients with solid tumors (prostate, hepatocellular, colon or renal cell), and demonstrated to be safe and well-tolerated [108].

On the other hand, the inhibition of L1CAM can be an interesting strategy to inhibit VCO. It is known that targeting the integrin–L1CAM signaling pathway can impair the development of metastases [41]. Many monoclonal antibodies can be used alone or in combination with chemotherapy to reduce proliferation [109]. Ab417, a monoclonal antibody against L1CAM has been proved to increase treatment response in intrahepatic cholangiocarcinoma in combination with chemotherapy [110] as well as reduce therapy-associated cardiotoxicity [111].

### 4.3. Cancer Cell Motility

One of the main characteristics of tumor cells that co-opt blood vessels is their increased invasiveness and migration capacity, which allows them to infiltrate the tissue structure [29]. Recent studies in different cancers have shown that cells that perform VCO behave differently from those that induce sprouting angiogenesis [14] but, in addition, this increased invasiveness is conditioned by several elements of the TME [29]. For example, bradykinin is a substance produced by brain endothelial cells, especially during tumor progression. Thus, the presence of this molecule in the microenvironment induces the chemotaxis of glioblastoma cells, which express the bradykinin receptor, making them attracted to the blood vessels to co-opt them [54]. We can also highlight the chemokine CXCL12 as a key element of the microenvironment produced by endothelial cells [54]. In a preclinical model of glioblastoma, it was shown that the application of antiangiogenic drugs promoted an increase in the expression of CXCR4 (CXCL12 receptor) in tumor cells [112]. Thus, cancer cells are attracted to the vessels to carry out VCO [112]. IL-8 has also been shown to increase the invasiveness of glioblastoma cells to develop VCO [113,114].

On the other hand, it has been recently discovered that cancer cell motility mediated by the Arp2/3 (actin-related protein 2/3) complex is necessary to promote VCO in liver metastases [17]. VCO requires the infiltration of tumor cells into the tissue so that tumors can develop around pre-existing blood vessels. Therefore, cell motility is considered as an essential mechanism for VCO.

The Arp 2/3 complex acts as a polymerization nucleus for actin at the apical domain of cells. Previous studies have discovered its involvement in tumor invasion in colorectal and breast cancer [17,115]. This complex is made up of seven subunits: ACTR2, ARP3, ARPC2, ARPC3, ARPC4 and ARPC5. In an in vivo study using an orthotopic mouse model of liver metastases originating from colorectal cancer, it was observed that silencing the ARPC3 subunit gene decreased the number of tumors that had a vascular cooptation growth pattern. These results suggest that cell motility plays a key role both in vessel co-option and in cancer cell invasion and progression, which raises the possibility of using the Arp2/3 complex as a therapeutic target to inhibit vascular cooptation [17].

The upstream of Arp2/3 complex, angiopoietin-1 (Ang1), also regulates cell motility. This protein is the main agonist ligand of the tyrosine kinase receptor Tie2 and its overexpression has been associated with processes such as metastasis and tumor angiogenesis. Moreover, Ang1 is known to be involved in VCO, although the molecular mechanisms involved are unknown. In a recent study, a correlation was found between Arp 2/3 expression levels in liver metastases and the presence of Ang1 in the tissue [115]. Rada et al. recently suggested that Ang1 could be a positive regulator of Arp 2/3 complex expression [116]. Ang1 allows cell interaction mainly through Tie2. Although this receptor is mainly expressed in endothelial cells and is relevant for vascular growth [117], it is also expressed in cancer cells and other cell types of the tumor microenvironment [118]. In the aforementioned study, Tie2 was found to mediate the expression of Arp 2/3 in liver metastases through Ang1 [116]. Tie2 is a receptor belonging to the PI3K/AKT (phosphoinositol 3-kinase/protein kinase B) signaling pathway, so the role of this pathway in Arp 2/3 expression was explored. It has been shown that the effect of Ang1 on the expression of the Arp 2/3 complex was reduced in the presence of LY294002, PI3K/AKT inhibitor [116].

#### Inhibition of Cancer Cell Migration for VCO Inhibition

Regarding the involvement of the Arp2/3 complex in cancer cell migration, its inhibition could be a therapeutic strategy to prevent VCO, as was demonstrated by Frentzas et al. [17]. These authors inhibited the expression of the ARP3C subunit of the Arp2/3 complex in HT29 cell lines by using shRNA, and they observed a reduction in cell migration. They also generated an experimental model of liver metastases using this cell line, and they observed a lower incidence of metastases with the rHGP growth pattern that undergo VCO. This confirms that the inhibition of cell motility through Arp2/3 complex prevents VCO [17]. Apart from this evidence, numerous inhibitors of Arp2/3 complex have been shown to reduce cancer cell migration and cancer progression [119,120]. It would be very interesting to study the effect of all these molecules in the context of VCO. CK-666 is a specific inhibitor of Arp2/3 complex that blocks the conformational change necessary to reach the native structure [119,120,121]. CK-666 has shown the ability to inhibit cancer cell migration and metastases in different studies using human glioma cells [122]. On the other hand, different CK-666 analogues have been shown to block B16-F13 melanoma cell migration. This effect has been observed in pancreatic ductal adenocarcinoma cells [121]. Benproperine is an FDA-approved drug used as an antitussive drug that has shown an inhibitory effect on cell migration and metastasis spread through Arp2/3 inactivation [123]. In vitro studies demonstrated that benproperine treatment significantly reduced colorectal, breast, melanoma, pancreas and prostate cancer cell migration [123]. Benproperine also reduced tumor growth and metastasis incidence in the orthotopic model of pancreatic ductal adenocarcinoma (PDAC) generated by AsPC-1 cells and the metastases growth in the spleen of colorectal cancer cells HCT-116 and DLD-1 [123,124]. Pimozide is an antipsychotic drug that has shown the ability to inhibit the migration and invasiveness of cancer cells from the colon, pancreas, lungs or melanoma [125]. These drugs or targeted gene silencing could be a very interesting strategy to study for the inhibition of VCO.

### 4.4. Epithelial-to-Mesenchymal Transition (EMT)

EMT is a process that epithelial cells undergo and that leads them to acquire the characteristics of mesenchymal cells. This is a phenomenon that occurs in normal physiological situations (embryogenesis, wound repair, fibrosis), but tumor cells can also experience it [24,29,126].

In cancer cells that undergo EMT, gene expression is modified by transcription factors, such as SNAIL, ZEB and bHLG among others, that suppress the expression of epithelial genes such as genes of proteins involved in tight junctions (ocludins, claudins, zonula ocludens) [126,127]. As a consequence of these changes on gene expression, tumor cells lose their polarity and cell–cell contacts are lost, increasing their migratory ability and invasiveness [24,126,128]. Moreover, these transcription factors induce the expression of mesenchymal genes such as N-cadherin or vimentin, which promote cancer cell invasiveness, cytoskeleton, cell mobility and focal adhesions [127]. During EMT process, tumor cells synthesize and secrete high amounts of collagens and fibronectin, which are the ligands for integrins. Integrin binding to the extracellular matrix components lead to cell signaling activation that promotes EMT [127]. Finally, cancer cells that undergo EMT produce proteases and MMPs that cleave extracellular matrix to favor cell migration [126].

In summary, EMT reduces cell adhesion between cancer cells and increases their mobility. All these together provide them the ability to invade the adjacent tissues. For this reason, EMT constitutes a mechanism by which cancer cells can access the blood vessels to co-opt them [24]. Although the molecular mechanism is not yet known in depth, in this section, we will describe previous studies that show a close relationship between EMT and VCO. Rada et al. analyzed gene expression by using RNA seq in these tissues and EMT-related genes were observed to be upregulated in metastases with replacement growth pattern (VCO) [129]. Moreover, E-cadherin immunohistochemistry (epithelial marker) revealed lower expression of this protein in metastases undergoing VCO than in metastases with angiogenic growth pattern [129].

In vitro studies with co-cultured hepatocytes and HT29 colorectal cancer cells showed higher vimentin expression suggesting that cancer cells are able to induce EMT in hepatocytes to promote replacement growth pattern [130].

EMT could be involved in the resistance of VCO tumors to anti-angiogenic therapy. Respecting this, Maione et al. detected an increase in mesenchymal markers in cancer cells of pancreatic neuroendocrine cancer and cervical cancer in mice treated with sunitinib [131]. Hammers et al. identified a reversible EMT process responsible for sunitinib resistance. These authors performed a preclinical model by using the PDX of the cutaneous metastasis of renal cell carcinoma. The patient had not responded to sunitinib treatment, but the tumor was responsive in the preclinical model, as there was a reduction in vascular density. An analysis of the cancer cell morphology determined that the patient had a spindle-shaped morphology (mesenchymal), whereas mice showed an epithelial morphology [132].

Malestein et al. demonstrated a similar phenomenon in response to sorafenib. Sorafenib-resistant cell lines showed higher levels of mesenchymal markers and lower levels of epithelial markers. These studies concluded that cancer cells undergo an EMT program in response to sorafenib [133]. Kucynski et al. performed an orthotopic model of hepatocellular carcinoma that was treated with sorafenib. Chronic treatment with sunitinib led to resistant tumors with an invasive behavior through VCO [22]. Transcriptomic analysis revealed an increase in gene expression related to EMT, allowing a higher invasiveness of cancer cells to co-opt liver capillaries [22]. Despite all these evidences, future work will be needed to unravel the importance and role of EMT in non-angiogenic tumor growth.

#### Inhibition of EMT as an Strategy for Vessel Co-Option Inhibition

Several molecules have been shown to inhibit EMT in different cancer types: pancreas, colorectal, breast, brain or lungs. In some studies, the inhibition of EMT has been associated with chemotherapy sensitivity, although its relationship with VCO has not yet been studied. In colorectal cancer, curcumin suppresses EMT and reverses 5-fluorouracil (5-FU) resistance. 5-FU resistant cells acquire mesenchymal characteristics. Curcumin sensitizes cancer cells to 5-FU treatment in HCT116 and SW480 cell lines, suppressing EMT and stemness [101]. In pancreatic cancer, mocetinostat, a class I HDAC inhibitor, interferes with ZEB1 function and induces sensitivity against chemotherapy in pancreatic cell lines [134]. Zidovudine sensitizes gemcitabine-resistant pancreatic cancer cells and inhibits EMT through the regulation of the Akt-GSK3b-Snail1 pathway [135]. Evodiamine decreases the expression of EMT-related genes such as Slug, Twist, Zeb1 and vimentin and sensitizes gastric cancer cells to oxaliplatin treatment [136].

Different molecules have been tested in breast cancer. Pyrvinum paomate, inhibits the Wnt pathway and suppress the expression of EMT markers such as Snail, N-cadherin or vimentin in breast cancer cells [137]. Palbociclib has been shown to inhibit EMT and metastasis in breast cancer cell lines MDA-MB-231 and T47D in a Jun/COX-2 dependent pathway [138]. In glioblastoma, Icaritin inhibits EMT in the glioblastoma (GBM) cell line U87MG in a PTEN/Akt/HIF-1α pathway [139]. Disulfiram inhibits TGF-β-induced EMT in MCF-7 and MDA-MB-231 breast cancer cell lines [140].

Several molecules have been shown to inhibit EMT in lung cancer. Moscatilin inhibits EMT in the human lung cancer cell line H460 and decreases the levels of survival pathways such as Erk and Akt [141]. On the other hand, Zerumbone and Bufalin inhibit TGF-β-induced EMT in A549 lung cancer cells [142,143]. Metformin has shown inhibitory EMT effects in ovarian cancer cells such as SKOV3 and A2780 [144].

Given that lung and brain tumors can develop VCO, as can breast and colorectal metastases, it is very interesting to test if all the mentioned molecules can inhibit VCO.

### 4.5. Extracellular Matrix and VCO

The extracellular matrix (ECM) is a large three-dimensional network of proteins and polysaccharides that provides structural support to cells and allows communication between them [81,145]. In animals, ECM is located forming sheet-like structures at the base of epithelial and endothelial cells and is known as the basement membrane, although it can also be located between cells [145]. In tumor tissues, the basement membrane is an essential component of tumor microenvironment and has a key role in VCO tumor growth, as it regulates the interaction between cancer cells and endothelial cells from pre-existing blood vessels. It has been described that collagen, laminin, fibronectin and vitronectin interact with adhesion molecules, such as integrins, that are expressed by cancer cells promoting their approach to the pre-existing blood vessels [41,44,146,147].

In most of the cases, VCO tumors compress pre-existing blood vessels causing deformations and reducing their functionality [51,148]. This fact decreases blood flow and generates hypoxic tumor cells in several regions. Lack of oxygen generates changes in ECM components: Hypoxia favors the formation of collagen I and regulates lysyl oxidase (LOX) enzyme that participates in its synthesis [149]. All these phenomena contribute to ECM stiffness that activates numerous signaling pathways in cancer cells involved in cell migration and vessel leakiness [51,150]. Moreover, it is accepted that LOX enzymes favor the colonization of new cancer cells in the lungs and breast [149,151,152]. On the other hand, a hypoxic environment also favors ECM remodeling through collagen degradation by MMPs that increase the invasiveness of cancer cells [149]. It has been shown that MMP2 and MMP4 are upregulated in colorectal cancer liver metastases [29].

On the other hand, the secretion of hypoxia inducible factor (HIF) promotes cancer-associated fibroblast (CAF) recruitment [149,153]. CAFs are cellular components of the stroma that contribute to an increase in ECM stiffness through the secretion of soluble and insoluble factors [153]. The presence of CAFs favors the migration and invasion of tumor cells through the production of collagen I and fibronectin [153,154,155]. Although a direct relationship of these cells with VCO has not been demonstrated, it is hypothesized that CAFs may be a factor contributing to the establishment and dissemination of tumor cells with this type of non-angiogenic growth [29].

Finally, ECM also acts as a source of cytokines and growth factors regulating their distribution and activation [29]. Changes in ECM induced by hypoxia favor the presence of immunosuppressive cells, such as M2 macrophages [149,150]. Moreover, the presence of CAFs and collagen I abundance in ECM promotes TGF-β liberation, which suppresses the immune system and increases collagen, fibronectin and proteoglycan abundance, promoting the migration and dissemination of cancer cells [149].

In summary, VCO growth induces relevant changes in the composition of the ECM that favors tumor progression. For this reason, understanding the role of the ECM in this type of growth may favor the development of targeted therapies to treat.

#### Inhibition of Extracellular Matrix as Strategy for VCO Inhibition

As it has been mentioned before, the changes in the arrangement of the different ECM components produced by tumor growth via VCO favor the progression of cancer cells. For this reason, there is a marked interest in the scientific community in developing specific inhibitors that can target certain molecules present in ECM. There are multiple strategies to inhibit ECM synthesis that can be used to target VCO. In the next paragraph, we will focus on the inhibition of the activity of two types of enzymes that are increased in tumors undergoing vessel co-option: LOX and MMP families.

The inhibition of the LOX family proteins involved in ECM remodeling has been postulated as an effective strategy against fibrotic and neoplastic diseases [81,156]. The levels of these enzymes are increased in most cancers and their increased levels may even be the cause of fibrosis in very aggressive cancers, since they favor collagen deposition. A large number of small molecules has been discovered that can specifically inhibit the activity of the LOX family of proteins. For example, the use of β-aminopropionitrile (βAPN) inhibits the metastatic capacity of breast cancer, cervical cancer, hepatocellular carcinoma and pancreatic cancer cells [156,157,158,159,160]. However, βAPN treatment has been discontinued due to toxicity problems. On the other hand, AB0023, an antibody that inhibits LOXL2 enzymatic activity to a greater extent than βAPN has also been developed [161]. However, its humanized version (simtuzumab) has not shown good results in combination with gemcitabine in clinical trials in patients with pancreatic and colorectal adenocarcinoma [81,162,163]. Despite this, many other molecules are being evaluated since it is thought that the use of LOX inhibitors could favor the action of other antitumor substances, such as chemotherapy.

Regarding LOX inhibition, it has been demonstrated that in an experimental model of metastatic breast cancer using 4T1 cells that undergo VCO [16], paclitaxel treatment enhance metastatic growth due to the deposition of ECM proteins by T cells. Very interestingly, the inhibition of LOX with βAPN eliminates the enhancement of metastatic growth. These results indicate that blocking ECM remodeling can eliminate cancer growth in vessel co-opting models [164].

MMP degrade the collagen of the ECM in hypoxic tissues, allowing cancer cells to migrate to other locations of the body. For this reason, their overexpression is associated with a worse prognosis [149]. Given that in non-angiogenic tumors a greater overexpression of MMP2 and MMP14 has been observed, it is of interest to inhibit these MMPs in order to reduce the aggressiveness of non-angiogenic tumor growth [29]. Specifically, the non-competitive inhibition of MMP2, MMP14 and MMP13 using potassium ferricyanide has shown in some studies significant reductions in tumor growth in a murine model of breast cancer lung metastases with VCO growth. However, the reduction was higher in combination with immunotherapy, thus enhancing the therapeutic effect of the anti-CTLA-4 antibody [165]. Therefore, it would be interesting to study the effect of this inhibitor in other non-angiogenic tumors.

It should be remarked that the increased expression of LOX and MMP enzymes occurs due to the lack of oxygen. This fact causes the absence of angiogenesis in co-opted tumors [149]. Therefore, some studies stress the relevance of using hypoxia-inducible factor (HIF) inhibitors not only to avoid these enzymes but also because HIF acts on other genes that encode proteins involved in cell survival, EMT, invasion, metastasis or resistance to antitumor agents. Many of the inhibitors have shown good preclinical results, while in clinical trials only two HIF-1α destabilizers, vorinostat and panobinostat, have shown effectiveness against multiple myeloma [166,167]. However, several HIF-1 inhibitors have shown better results in combination with other therapeutic agents. For this reason, more preclinical and clinical studies are essential to truly evaluate the potential of this potential combination therapy.

## 5. Conclusions and Future Perspectives

Although the knowledge of the existence of VCO tumors has been established for several decades [23], there are many open questions that researchers and clinicians need to address in the next years.

One of the most important contributions of the understanding of the non-angiogenic mechanisms of vascularization is the change of paradigm of the tumor angiogenesis and the role of VEGF as the key element for tumor vascularization, described by Folkman [76]. The new conception of alternative mechanisms of vascularization such as VCO or vascular mimicry has given the explanation to scientists and clinicians to answer why numerous cancers do not respond to AAT. However, most of the molecular features that can explain the relationship between VCO and ATT need to be studied in the future. Do all cells vessel co-opting tumor cells produce VEGF? Do co-opted blood vessels express VEGFR2?

Tumor microenvironment components in lungs, liver or brain need to be studied when the tumors or metastases that are colonizing these tissues are undergoing VCO. Firstly, blood flow needs to be assessed in all the different VCO growth patterns and in different histology subtypes. It could be guessed that co-opted vessels that are quiescent and mature can provide tumors with adequate blood flow, but the proliferation of cancer cells that are co-opting blood vessels can also compress them, reducing blood flow. Understanding this phenomenon can encourage the design of new strategies based on the use of vasodilators. The reduction of blood flow can determine: hypoxia levels, delivery of therapeutic agents and changes in tumor microenvironment and immune cell infiltration.

Hypoxia can mediate numerous resistance mechanisms in tumors that are being treated with chemotherapy or immunotherapy. Moreover, hypoxia can promote cancer cell EMT, invasiveness and metastases. On the other hand, inadequate drug delivery is a very important limitation of cancer treatment that needs to be overcome. On the other hand, changes in tumor immune microenvironment need to be studied to assess the administration of different immunotherapies in these VCO tumors. It has been described that liver metastases from colorectal cancer undergoing angiogenesis show higher CD8+ levels. Moreover, VCO is associated with the presence of M1 macrophages and matrix remodeling macrophages in the TME [38]. Neutrophils appear to correlate positively with VCO [28].

In this review, we have focused on the different cellular mechanisms involved in VCO that can be inhibited by the different molecules already described in cancer. However, most of these studies have been performed without studying the effects on tumor vascularization. We believe that most of these inhibitors of cell adhesion, EMT, ECM, cell migration or even blood vessel modulators are very good tools for VCO inhibition.

## Figures and Tables

**Figure 1 ijms-25-00921-f001:**
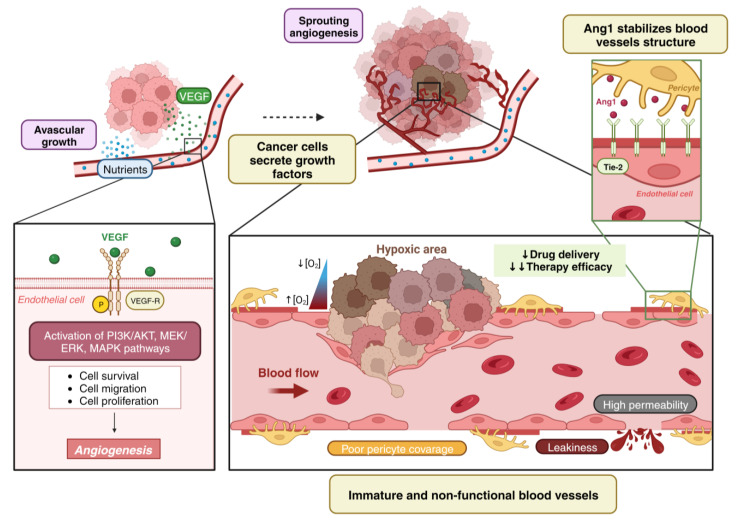
Tumor angiogenesis leads to non-functional blood vessels. Small tumors have an avascular growth, receiving nutrients and oxygen by diffusion. When they reach a certain size and the levels of pro-angiogenic factors (such as VEGF) are higher than the anti-angiogenic factors, an angiogenic switch takes place. VEGF signaling through its receptor VEGFR2 promotes endothelial cell proliferation, migration and survival to by the activation of multiple pathways, which lead to angiogenesis. Tumor blood vessels are poorly covered by pericytes and leaky leading to increased hypoxia. Hypoxia in solid tumors is also generated for the rapid growth of tumors that increase the oxygen demand that cannot be supplied [13]. Pericytes secrete Ang1, which binds with the Tie-2 receptor in endothelial cells. Ang 1 promotes the interaction between endothelial cells and pericytes to stabilize the new vessels’ structure. Nevertheless, the tumor vessels are not mature because the pericyte coverage is partial. Consequently, they have high permeability and are not functional resulting in a poor drug delivery and ineffective therapies. Created with BioRender.com, accessed on 8 January 2024.

**Figure 2 ijms-25-00921-f002:**
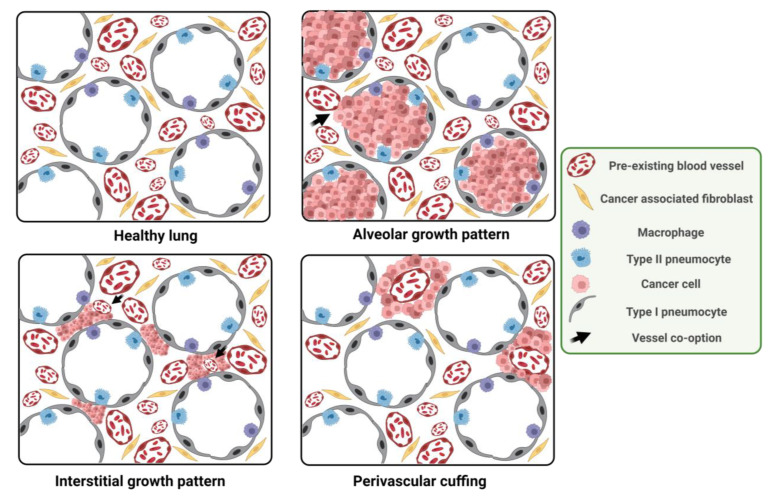
Histological features of VCO vs. angiogenic tumors in the lungs. Healthy lung show type I and type II pneumocytes covering each alveoli. In the VCO alveolar growth pattern cancer cells grow to fill air spaces inside the alveoli. Lung parenchyma is preserved and pneumocytes and pre-existing blood vessels are in the same interface. In the VCO interstitial growth pattern, cancer cell growth in the alveolar wall compresses air spaces. In the perivascular cuffing growth pattern, cancer cells grow surrounding pulmonary big vessels. Schematics have been made using BioRender.com accessed on 8 January 2024.

**Figure 3 ijms-25-00921-f003:**
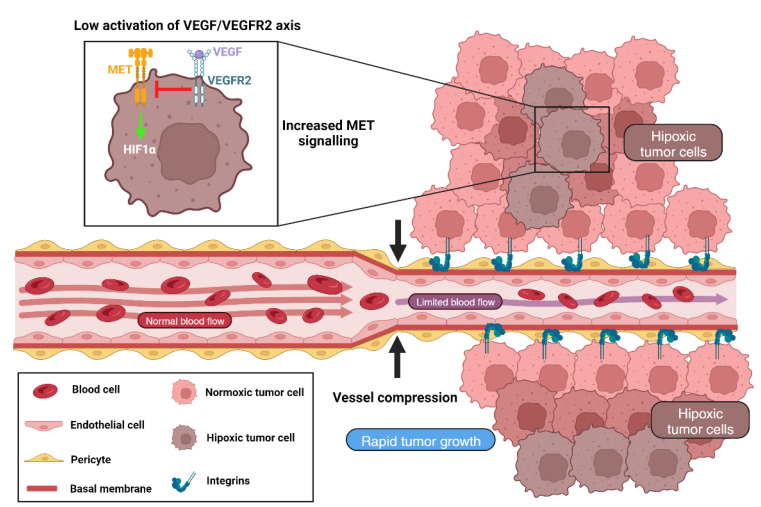
VCO induces tumor hypoxia. Different mechanisms in VCO induce tumor hypoxia. During VCO, cancer cells hijack pre-existing blood vessels and compress them. This generates a decrease of blood flow that leads to hypoxia. At the same time, cancer cell proliferation increases oxygen demand, that the reduced blood flow cannot supply. This lead to a decrease of the oxygen levels in the TME. On the other hand, during VCO, the VEGF/VEGFR2 signaling pathway is downregulated, and this downregulation lead to an increase of MET signaling. MET activation activates hypoxia inducible factors, such as HIF1α.

**Figure 4 ijms-25-00921-f004:**
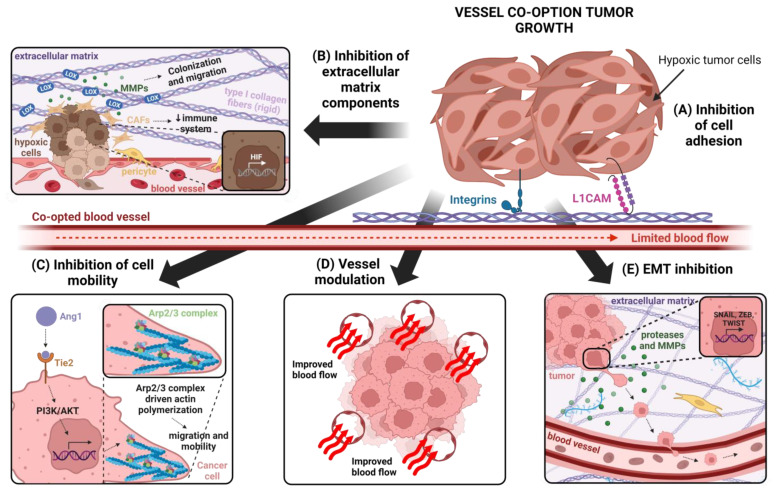
Different strategies for VCO inhibition. (**A**) Inhibition of the adhesion between cancer cells and pre-existing blood vessels. Cancer cells growing by VCO are adhered to pre-existing blood vessels thanks to the participation of α1, α3 and β1 integrins or the L1CAM glycoprotein. For this reason, different molecules and antibodies that act as inhibitors and prevent tumor cell attachment to pre-existing blood vessels can be used to inhibit VCO. (**B**) Inhibition of ECM components. The compression of tumor blood vessels limits blood flow and oxygen distribution in the tumor microenvironment. Consequently, hypoxia induces changes in ECM since it favors the formation of LOX enzymes and MMPs involved in the colonization of new tumor cells and cell migration, respectively. In addition, secretion of HIF promotes the recruitment of CAFs that increase the stiffness of the extracellular matrix and favor tumor cell invasion. Finally, oxygen deprivation contributes to the release of immunosuppressive cells such as M2 macrophages and TGF-β that reduce the efficacy of the immune system. Thus, the study of inhibitors of HIF, LOX and MMPs (MMP2 and MMP14) may be an effective strategy to combat the harms of vascular co-opted tumors. (**C**) Inhibition of cell motility. In liver metastases with VCO, it has been shown that the Arp 2/3 complex promotes tumor cell migration by polymerization of actin filaments. Therefore, it is of interest to study the involvement of this complex in this type of tumor growth in other cancers and to find effective inhibitors to prevent VCO, since the use of shRNA has shown efficacy in experimental models with the HT29 cell line. (**D**) Modulation of tumor blood vessels. The inadequate blood flow that reaches tumor cells favors the generation of a hypoxic environment that promotes cancer cells to acquire an aggressive phenotype. For this reason, efforts are being made to develop therapies (vascular normalization and vascular promotion) that improve the functionality of tumor blood vessels and induce the generation of new vessels through the process of angiogenesis to increase the delivery of oxygen and drugs to the tumor, thus improving the efficacy of treatments. (**E**) Inhibition of epithelial–mesenchymal transition. It has been demonstrated the increased expression of EMT-related genes in colorectal metastases that co-opt tumor blood vessels. Furthermore, the use of anti-angiogenic agents in VCO tumors results in increased tumor invasiveness due to a shift of the cells towards a mesenchymal phenotype. Therefore, the use of inhibitors against genes involved in EMT transition such as SNAIL, ZEB and TWIST have shown efficacy in lung, brain, colorectal and breast tumors that can develop VCO growth. Figure created using BioRender.com accessed on 8 January 2024.

**Table 1 ijms-25-00921-t001:** Molecules of interest for VCO inhibition.

	Molecule	Effect
Anti-angiogenic agents	Aflibercept	Anti-VEGF glicoprotein
	Bevacizumab	Anti-VEGF monoclonal antibody
	Cabozantinib	Inhibitor of tyrosine kinase domain VEGFR
	Cediranib	Inhibitor of tyrosine kinase domain of VEGFR1, VEGFR2 and VEGFR3
	Lenvatinib	Inhibitor of tyrosine kinase domain of VEGFR1, VEGFR2 and VEGFR3
	Ramucirumab	Anti-VEGFR2 monoclonal antibody
	Regorafenib	Inhibitor of tyrosine kinase domain of VEGFR1, VEGFR2, VEGFR3 and TIE2
	Sorafenib	Inhibitor of tyrosine kinase domain of VEGFR2
	Sunitinib	Inhibitor of tyrosine kinase domain of VEGFR2
Pro-angiogenic agents	Vandetanib	Inhibitor of tyrosine kinase domain of VEGFR2
	Eribulin	Microtubule dynamics inhibitor
	ldCil	Increase levels of VEGFR2
	Lysophosphatidic acid (LPA)	Tightens endothelial cell contacts
Inhibitors of cancer cell–blood vessel adhesion	Ab417	L1CAM antibody
	AIIB2	β1 integrin inhibitor
	ATN-161	β1 integrin inhibitor
	OS2966	β1 integrin inhibitor
Inhibitors of cancer cell migration	Benproperine	Inhibitor of Arp2/3 complex
	CK-666	Inhibitor of Arp2/3 complex
	LY294002	Inhibitor of PI3K/AKT
	Pimozide	Inhibitor of Arp2/3 complex
	Bufalin	Inhibits TGF-β-induced EMT
Inhibitors of EMT	Curcumin	Inhibitor of EMT in 5-FU resistant cells
	Disulfiram	Inhibits TGF-β pathway
	Evodiamine	Decrease the expression of Slug, Twist, Zeb1 and vimentin.
	Icaritin	Inhibits PTEN/Akt/HIF-1α pathway
	Metformin	Inhibits EMT
	Mocetinostat	Interfere with ZEB1 function
	Moscatilin	Inhibits Erk and Akt pathways
	Palbociclib	Regulates Jun/COX-2 pathway
	Pyrvinum paomate	Inhibits the Wnt pathway
	Zerumbone	Inhibits TGF-β-induced EMT
	Zidovudine	Inhibits Akt-GSK3b-Snail1 pathway
ECM regulators	AB0023	LOXL2 inhibitor
	βAPN	LOXL2 inhibitor
	MMPs	Collagen degradation
	Simtuzumab	Anti-LOXL2 monoclonal antibody
Hypoxia inhibitors	Panobinostat	Inhibitor of HIF-1α
	Vorinostat	Inhibitor of HIF-1α

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
