# Peer review of "The Inhibition of Vessel Co-Option as an Emerging Strategy for Cancer Therapy"

_ijms, 2024, doi:10.3390/ijms25020921_

Round 1

Reviewer 1 Report

Comments and Suggestions for Authors

Q1:Figure 1 Tumor blood vessels are poorly covered by pericytes and leaky leading to increased hypoxia.

Dose this sentence suggests that hypoxia is worsened because of the leaky? How about “Hypoxia is a common feature of solid tumors, and develops because of the rapid growth of the tumor that outstrips the oxygen supply” (Please see Cancer Cell Int. 2021 Jan 20;21(1):62. doi: 10.1186/s12935-020-01719-5. For further information)

Q2: Line 80: displacing non-tumoral cells and recruiting” immune cells”

Please clearly define “immune cells” you mentioned in your article

Q3: non angiogenic?   Non-angiogenic

Q4: both types of vascularization? what are them?

Q5:line 98-100  In tumors undergoing VCO, anti-tumor immunity is compromised by  different mechanisms than in angiogenic tumors, being these mechanisms not described to date

Are you sure your comments which were summarized from your references? I spent some times in reading your reference 2, for example, In their figure 1, Icannot find anything regarding your” compromised anti-tumor immunity “ ?

Q6: In this paper, you mentioned several times on hypoxia, HIF, and HRE. However, there are still several molecules you should introduce to your future readers.

Please see the Front Oncol. 2022; 12: 965277. doi: 10.3389/fonc.2022.965277

Their figure 3, figure 4, and Figure 5 are very clear. Please summarize this paper and systemize a distinct figure to help your future readers to understand those molecular cascades related VCO.

Author Response

Q1: Figure 1 Tumor blood vessels are poorly covered by pericytes and leaky leading to increased hypoxia.

Dose this sentence suggests that hypoxia is worsened because of the leaky? How about “Hypoxia is a common feature of solid tumors, and develops because of the rapid growth of the tumor that outstrips the oxygen supply” (Please see Cancer Cell Int. 2021 Jan 20;21(1):62. doi: 10.1186/s12935-020-01719-5. For further information)

We thank the reviewer for pointing this out. According to the reviewer suggestion we have added this sentence in current line 74: Hypoxia in solid tumors is also generated for the rapid growth of tumors that increase the oxygen demand that can not be supplied

Q2: Line 80: displacing non-tumoral cells and recruiting” immune cells’. Please clearly define “immune cells” you mentioned in your article

We thank the reviewer for asking us to clearly define ‘immune cells’. As the recent literature says, immune cell population in vessel co-option can be different depending on the tissue. Teuwen et al., (2021) show the presence of M1 or matrix remodeling macrophages in lung metastases. On the other hand, Peter Metrakos research team study the role of neutrophils in vessel co-option in different papers (Rada et a., 2022). For this reason, we have modified the sentence as follows: cells and recruiting immune cells such as neutrophils or M1 macrophages

Q3: non angiogenic?   Non-angiogenic

We thank the reviewer for pointing this out. In the new version, we have amended this minor mistake.

Q4: both types of vascularization? what are them?

We referred to angiogenesis and vessel co-option. We have added this in current line 100-101

Q5: line 98-100  In tumors undergoing VCO, anti-tumor immunity is compromised by  different mechanisms than in angiogenic tumors, being these mechanisms not described to date

Are you sure your comments which were summarized from your references? I spent some times in reading your reference 2, for example, In their figure 1, I cannot find anything regarding your” compromised anti-tumor immunity “ ?

We have removed this sentence, as the reference 2 is not saying this. We thank the reviewer for pointing this out.

Q6: In this paper, you mentioned several times on hypoxia, HIF, and HRE. However, there are still several molecules you should introduce to your future readers.

Please see the Front Oncol. 2022; 12: 965277. doi: 10.3389/fonc.2022.965277

Their figure 3, figure 4, and Figure 5 are very clear. Please summarize this paper and systemize a distinct figure to help your future readers to understand those molecular cascades related VCO.

We thank the reviewer for this suggestion. We have added a new paragraph in line 462, and we have created a new figure (Figure 3) summarizing the mechanisms of VCO regulating tumor hypoxia. We think that this improves our manuscript

Reviewer 2 Report

Comments and Suggestions for Authors

This paper addresses molecular mechanisms involved in vessel co-option (VCO). It is helpful for understanding the existing techniques for VCO inhibition which is important for searching new approaches to treatment of colorectal, breast and other cancers. I recommend a few revisions before publication of this manuscript.

1. You are right stating that "microvascular growth patterns can vary in time and space". Besides temporal growth patterns, which are different for angiogenesis and VCO, what about spatial microvessel patterns?  Do they also differ? With respect to review status of this paper, it makes sense to address this issue and discuss applicability of vessel pattern analysis to VCO detection on histological or/and dermatological scales (see e.g. 10.3892/ol.2019.10070,  10.1016/j.adengl.2012.06.007, 10.1364/BOE.420786, etc.)

2. In Discussion, Figure 3 seems insufficient to compare different strategies for VCO inhibition. I miss the table summarizing both qualitative and quantitative features of these approaches: what molecules and/or stressors are necessary, how long the whole procedure lasts, what equipment is involved, etc.

3. Please double-check the manuscript and correct typos like "componentes" (should be "components"), "non angiogenic" - (should be "non-angiogenic"), etc.

Comments on the Quality of English Language

Moderate English style and spell check is necessary.

Author Response

This paper addresses molecular mechanisms involved in vessel co-option (VCO). It is helpful for understanding the existing techniques for VCO inhibition which is important for searching new approaches to treatment of colorectal, breast and other cancers. I recommend a few revisions before publication of this manuscript.

We thank reviewer 2 for his/her suggestions and for supporting the publication of this manuscript.

  1. You are right stating that "microvascular growth patterns can vary in time and space". Besides temporal growth patterns, which are different for angiogenesis and VCO, what about spatial microvessel patterns?  Do they also differ? With respect to review status of this paper, it makes sense to address this issue and discuss applicability of vessel pattern analysis to VCO detection on histological or/and dermatological scales (see e.g. 10.3892/ol.2019.10070,  10.1016/j.adengl.2012.06.007, 10.1364/BOE.420786, etc.)

We thank the reviewer for this very interesting suggestion. However, this review is focused on highly vascularized tissues such as lung, liver or brains where VCO occurs frequently. However, although we consider very interesting these suggestions, we believe that changing our focus to skin cancers does not contribute improvements to our article.

  1. In Discussion, Figure 3 seems insufficient to compare different strategies for VCO inhibition. I miss the table summarizing both qualitative and quantitative features of these approaches: what molecules and/or stressors are necessary, how long the whole procedure lasts, what equipment is involved, etc.

According to the reviewer suggestion, we provided now a table with all the molecules that we propose for VCO inhibition. However, there are some aspects that we have not added because there are not enough information until now (Schedule of the treatments and equipment).

  1. Please double-check the manuscript and correct typos like "componentes" (should be "components"), "non angiogenic" - (should be "non-angiogenic"), etc.

We have checked all this typos. Thank you very much for pointing this out.

Yours sincerely,

José M. Muñoz-Félix